# Prognostic Implications of Epilepsy Onset Age According to Relapse Pattern in Patients with Four-Year Remission

**DOI:** 10.3390/diagnostics10121089

**Published:** 2020-12-14

**Authors:** Soochul Park, Myeongjee Lee

**Affiliations:** 1Department of Neurology, College of Medicine, Yonsei University, Seoul 03722, Korea; 2Biostatistics Collaboration Unit, Department of Biomedical Systems Informatics, College of Medicine, Yonsei University, Seoul 03722, Korea; lmj7914@gmail.com

**Keywords:** epilepsy, age at onset of epilepsy, remission within 1 year, relapse pattern, prognostic factor

## Abstract

A total of 472 epilepsy patients with a 4-year remission period were divided into 10-year age groups according to age of onset. The relapse patterns during at least 3 years of follow-up were classified as early relapse (ER), late relapse (LR), and seizure-free (SF). The remission probability and multiplicity of prognostic factors were evaluated using univariate and multivariate multinomial logistic analyses. The weighted risk score based on odd ratios (ORs) was used for comparisons of the relative risk of relapse between groups. The group with onset in their 20s had the lowest remission probability among the groups. The risks of relapse in the LR patients and the relative weighted risk score of ER patients in the group with onset in their 20s were 3.11 and 19.44, respectively, which was the highest risk among the age groups. Patients without remission within 1 year had the highest relapse risk, with an OR of 7.18 in ER patients. The OR of relapse in patients with >10 generalized tonic–clonic (GTC) seizures was the second most important prognostic factor in LR patients. The distinct risk and corresponding prognostic factors in LR and ER patients reflected inherent differences between these relapse patterns.

## 1. Introduction

Age at onset of epilepsy and symptom duration have been consistently reported as important prognostic factors. The onset age of this disease varies due to its multifaceted etiology and inevitably affects symptom duration. Thus, the age at onset of epilepsy and symptom duration are closely correlated with each other. It has long been known that the response to treatment in newly diagnosed epilepsy patients is better than that in chronic epilepsy patients [1]. Thus, early effective treatment may be important for preventing progression to chronic epilepsy. Localization-related seizures, relevant lesions on a brain scan, achievement of seizure control within 1 year, and the total number of generalized tonic–clonic (GTC) or generalized seizures have been reported as important prognostic factors, in addition to conventional prognostic factors such as antecedents of epilepsy. High initial seizure frequency before the initiation of antiepileptic drugs (AEDs) and being a non-responder without achievement of seizure control within one year have also been reported as factors predictive of a poor prognosis [2,3,4,5,6]. In contrast, some reports [7,8,9] have shown that early intervention with AED therapy did not alter the long-term prognosis, leading to debate on this topic. Incomplete achievement of remission or relapse back to the pretreatment condition result in the prolongation of the total symptom duration. These results suggested the importance of achieving early control of seizures. Therefore, almost all prognostic factors are related to symptom duration.

Although age at onset of epilepsy and symptom duration have been consistently reported as important prognostic factors, studies assessing these factors in detail according to onset age have not yet been performed. We aimed to elucidate the prognostic implications of the age of onset stratified by 10-year increments with regard to relapse patterns among patients with 4 seizure-free years.

## 2. Materials and Methods

### 2.1. Patient Registration and Inclusion Criteria

A previous study [10] on relapse after achieving and maintaining complete seizure control for at least four years was continued after the endpoint of the study in July 2014. A total of 525 epilepsy patients were consecutively recruited from March 1999 to June 2017 to complete three more years of follow-up. Thirty-three patients were lost to follow-up after the initiation of AED withdrawal, and another 20 patients were excluded for various reasons before the initiation of AED withdrawal. Patients who had undergone epilepsy surgery, patients with juvenile myoclonic epilepsy (JME)—who exhibit a greater tendency towards relapse—and patients with benign childhood epilepsy with centrotemporal spikes were excluded. The presence of epileptiform discharges during the electroencephalography examination immediately before the initiation of AED withdrawal was another exclusion criterion. Three patients who exhibited a complex partial seizure as pure ictal amnesia only, as ascertained during a period of remission or later, were excluded. Finally, 472 patients were enrolled (Figure 1). All of the patients were regularly treated and followed by the corresponding author.

### 2.2. Patient Grouping and Follow-Up

The enrolled patients were divided into 5 groups according to seizure onset age stratified by 10-year increments as follows: under 10 years of age, 20–29 years, 30–39 years, 40–49 years, and 50 years or older. Each of the age groups was further stratified based on the relapse patterns that were investigated in the previous study [10], namely, early relapse (ER), late relapse (LR), and seizure-free (SF). SF patients were followed for at least 3 years.

### 2.3. AED Treatment and Policy Regarding AED Withdrawal

In principle, alternative monotherapy was attempted for the patients who had not achieved seizure freedom with the initial monotherapy. However, considering the patient’s clinical status, an add-on combination trial was attempted to shorten the time to achieve seizure control. Therapeutic drug monitoring of AEDs was regularly performed to guide adjustment of the dose and monitor compliance. Phenytoin, carbamazepine, valproate, lamotrigine, topiramate, and levetiracetam were used. In a few patients who had been referred, phenobarbital was administered. Planned AED withdrawal was performed as described in a previous study [10]. In the case of patients taking levetiracetam, the usual prescribed dose was 750 to 2500 mg daily, which was gradually reduced by 250–500 mg at every visit.

### 2.4. Standard Protocol Approvals and Patient Consent

The study design (Institution Review Board (IRB) No. 2020-1333-001) was approved by and adhered to the guidelines established by our hospital’s Human Research Protection Center. The need for informed consent was waived because of the retrospective nature of this study.

### 2.5. Data Analysis

The demographic data included sex, mean age of epilepsy onset, and mean symptom duration (duration > 120 months). The prognostic variables were an antecedent of epilepsy, no remission within one year after the initiation of AED treatment, >10 generalized seizures or 2 GTC seizures before the achievement of remission, general characteristics of the seizures such as epilepsy type (localization-related, idiopathic generalized, or undifferentiated epilepsy), relevant lesions on brain magnetic resonance imaging (MRI) scans, and nocturnal preponderance. The demographic data are also presented according to the relapse pattern within each age group (Table 1 and Table 2). Total symptom duration was defined as the sum of the duration from seizure onset to registration at the epilepsy clinic and the duration from registration to the most recent seizure attack. The total duration of the SF period was divided into the duration from the most recent seizure attack to the initiation of AED withdrawal and the duration of the AED withdrawal process (Appendix A).

Perinatal insults, febrile convulsions, a family history of epilepsy, head trauma combined with loss of consciousness for >1 h, and a history of infections of the central nervous system were regarded as antecedents related to epilepsy. The presence of at least one of these antecedents was regarded as the presence of antecedents. Nocturnal preponderance was defined as the occurrence of >90% of habitual seizures during sleep. Localization-related epilepsy, including remote symptomatic epilepsy, was based on the partial feature of ictal semiology, localized epileptiform discharge in the regular follow-up of the electroencephalogram, which was performed at intervals of at least two years, and relevant lesions in a brain MRI scan, defined according to the criteria for epilepsy and epileptic syndrome proposed by the International League Against Epilepsy (ILAE) [11]. Cryptogenic epilepsy was included in localization-related epilepsy and undifferentiated patients were treated as missing values statistically.

The mean OR of each prognostic factor in each onset age group was defined as the weighted risk score for the evaluation of the important prognostic factors across onset age groups. However, the relative weighted risk score based on the risk factors in SF patients as a reference was obtained as follows: the mean ORs for each variable in LR and ER patients were divided by the mean OR of the corresponding variable in SF patients in each age group, and the sums in each age group, obtained from the above calculation, were divided by five, which was the total number of prognostic factors. The mean of the multiplicity of the prognostic factors in each age group was used to evaluate the tendency and the proportion of factors predictive of a poor prognosis in each age group.

### 2.6. Statistical Analysis

Kaplan–Meier curves for each onset age group were compared via the log-rank test and Bonferroni adjustment. Important prognostic factors and the scores involving the multiplicity of the prognostic factors in each onset age group with regard to the relapse patterns were assessed. Multicollinearity among the variables was determined before the logistic regression analysis was performed. A univariate multinomial logistic regression model was constructed, and the baseline level for each variable was based on a favorable prognosis. The risk factors with a *p*-value <0.01 in the univariate model were included in the multivariate multinomial logistic regression model. Multivariate multinomial logistic regression was performed for LR and ER patients separately, with SF patients used as a reference. ORs with 95% confidence intervals (CIs) were calculated.

## 3. Results

### 3.1. Demographic Characteristics

A total of 472 patients were enrolled, with 9.1% (43 patients), 40.7% (192 patients), 18% (85 patients), 11.2% (53 patients), 11.6% (55 patients), and 9.3% (44 patients) assigned to the following groups, respectively: under 10 years old, 20–29 years old, 30–39 years old, 40–49 years old, and 50 years old or older (Table 1 and Table 2). The mean ages at seizure onset in these groups were 5.2 ± 2.8, 14.2 ± 2.4, 23.8 ± 2.8, 34.6 ± 2.9, 44.7 ± 3.2, and 57.9 ± 6.4 years, respectively, which were all significantly different (*p* < 0.0001) (more details are presented stratified by a relapse pattern in Table 1 and Table 2). There were no statistically significant differences in the mean age at seizure onset according to the relapse pattern within each age group. The mean duration of symptoms in all patients was 123.72 ± 113.16 months, and that in each onset age group was 233.3 ± 124.1, 154.0 ± 113.6, 120.6 ± 97.0, 77.8 ± 92.2, 51.9 ± 55.8, and 35.6 ± 44.6 months, respectively, which were significantly different (*p* < 0.0001). A significant difference in mean symptom duration stratified by relapse patterns was also noted between the groups that were 10–19 and 20–29 years old. There was a significant relationship between age at onset of epilepsy and symptom duration in all subjects (correlation = 0.47, *p*-value < 0.001; ß = −0.06, *p*-value < 0.001) (Figure 2).

In all patients, the mean duration of the SF period was 73.19 ± 24.12 months, the mean duration from the most recent seizure attack to the initiation of AED withdrawal was 59.91 ± 21.33 months, and the mean duration of the AED withdrawal process was 15.45 ± 9.19 months. There were no significant differences between the onset age groups or between the subgroups stratified by relapse patterns (Appendix A).

### 3.2. Statistical Data

The overall relapse rate was 63.5% (299 of 472 patients), and the proportions of patients with relapse among LR and ER patients were 38.5% and 24.8%, respectively. The group with onset in their 20s had the worst remission probability according to the Kaplan–Meier curve, and the groups with onset after age 40 had a significantly better prognosis than those with onset before age 30 according to the log-rank test with Bonferroni adjustment (Figure 3).

The variance inflation factors from the multicollinearity tests among the variables showed no correlations (<0.8) (Table 2). Among LR patients, a symptom duration >120 months (OR: 1.76, 95% CI: 1.13–2.70) and >10 GTC seizures (OR: 2.05, 95% CI: 1.29–3.67 and 4.30, 2.58–7.15) were found to be significant covariates in the univariate analysis. The significant covariates in ER patients were a symptom duration >120 months (OR: 4.31, 95% CI: 2.62–7.11), no remission within 1 year (OR: 9.98, 95% CI: 5.29–18.83), >10 GTC seizures (OR: 4.30, 95% CI: 2.58–7.15), and localization-related epilepsy (OR: 2.68, 95% CI: 1.26–5.70). The presence of relevant lesions on brain MRI scans was not a significant covariate but was included in the multivariable regression analysis (Table 3).

Regarding the weighted risk score, no remission within 1 year was the most important prognostic factor in every age group in ER patients, while in SF and LR patients, localization-related epilepsy was the most significant factor in every age group, except for the group with onset before age 10 (Figure 4).

The weighted risk score for almost all prognostic factors in every age group was higher in ER patients than in SF and LR patients (Figure 4). The number of patients with any combination of prognostic factors was highest in the group with onset between 10 and 19 years old, regardless of relapse pattern (details not provided). Patients with three prognostic factors accounted for the largest proportion, namely, 7 (53.8%) of 13 patients in the group with onset under 10 years old had a symptom duration >120 months, no remission within 1 year, and >10 GTC seizures. In the group with onset from 10–19 years old, the combination of a symptom duration >120 months, >10 GTC seizures, and localization-related epilepsy accounted for the largest proportion of patients, namely, 10 (27.0%) of 37 patients. In patients whose onset ages were 20–29 years old, 30–39 years old, and 40–49 years old, localization-related epilepsy and relevant lesions on brain MRI scans were identified in 19 (55.9%) of 34, 20 (76.9%) of 29, and 15 (75%) of 20 patients, respectively. However, in those 50 years old and older, localization-related epilepsy was the most important prognostic factor and was present in 13 (52%) of 25 patients (Figure 5a) (percentages not shown). A two-way ANOVA of the mean multiplicity of prognostic factors in each age group across relapse patterns indicated a marginal significance (*p* = 0.08) with a decreasing tendency, and the results in the LR and ER patients showed significant decreasing tendencies (Figure 5b). In the multivariate multinomial logistic regression analysis with SF patients as the reference (Table 4), the risk of relapse in LR patients was 3.11-fold higher in the group with onset in their 20s than in any other group. There were no significant differences among any of ER patient onset age groups. However, all the relative weighted risk scores for ER patients were higher than those for LR patients. The score for ER group with onset in their 20s was the highest, at 19.44, while that of the LR group with onset in their 20s was 3.22. The risk of relapse for LR patients with >10 GTC seizures was 1.88-fold higher than that for patients without this risk factor, and the remaining covariates in LR patients were not significant. In ER patients, the risk of relapse in patients without remission within 1 year was 7.18-fold higher (the highest risk) than that in patients without this risk factor. The risk of relapse in patients with >10 GTC seizures, localization-related epilepsy, and relevant lesions on brain MRI scans was 2.82-, 2.64-, and 1.83-fold higher in ER patients than in SF patients as a reference after statistical correction for the other covariates.

## 4. Discussion

Symptom duration is a well-known prognostic factor that is inevitably related to the age at onset of epilepsy. In addition to the conventional prognostic factors, such as relevant lesions on brain MRI scans and seizure classification, other prognostic factors, such as achievement of seizure control and the total number of generalized or GTC seizures, could be related to the symptom duration. Similar to the intercorrelation between age at onset of epilepsy and symptom duration, seizure control within the first year after the initiation of AED treatment and the total number of generalized or GTC seizures until the achievement of remission have been found to be temporally correlated with each other in the early phase of epilepsy, possibly influencing symptom duration and prognosis. Many studies have regarded the achievement of 12-month seizure control or freedom from seizures as a standard indicator of a therapeutic response, which can be used for the prediction of prognosis after the second AED trial [12]. The frequency and total number of seizures before or during AED treatment were identified as significant risk factors for seizure relapse [5,13] and were also important for the early control of seizures as clinical predictors of resistance to drug therapy [2]. The occurrence of many seizures before therapy and an insufficient response to initial AED treatment are indicators of refractory epilepsy. Similarly, the response to the first AED was the strongest predictor of long-term prognosis in adults and children [13,14,15,16,17,18]. In patients with childhood temporal lobe epilepsy (TLE), failure of the first AED trial accurately predicted refractory TLE at 2 years after the onset of seizures [15].

The overall relapse rate in this longitudinal study and the proportion of patients with relapse in the LR and ER groups were not substantially different from those reported in a previous study [10]. These results are in accordance with previous findings [19], indicating that overall outcomes in patients with newly diagnosed epilepsy have not markedly improved, even though many new AEDs with different mechanisms of action have been introduced. These findings may reflect the inherent nature of this disease entity. In terms of exclusion criteria of the recruitment process, JME, which has been presumed to be a lifelong genetic trait with a well-known tendency to relapse after withdrawal of AEDs. So, the majority of patients with JME need prolonged AED treatment after a follow-up of two decades and should be processed separately for the prognostic study. In this study, one patient was recognized before AED withdrawal and excluded, which had not an effect on the relapse rate.

A longer symptom duration does not necessarily indicate a worse prognosis. Age at onset of epilepsy and symptom duration were found to be significantly inversely related in this study, meaning that the symptom duration in patients with a younger age of onset was longer than that in patients with an older age of onset. However, the significance seemed to be limited, as the results from the multicollinearity test performed to rule out the possibility of intercorrelation among the variables indicated that there were negative relationships between symptom duration and the other variables, including onset age. The remission probability was the highest in the group with onset after age 50, which was used as a reference for the multinomial regression model. The multiplicity in the prognostic factors in each age group and the mean multiplicity of prognostic factors showed a decreasing tendency with increasing age, regardless of relapse pattern. In addition, the multivariate multinomial logistic regression analysis with SF patients as a reference and statistical correction for the other covariates demonstrated that a symptom duration >120 months was not a significant prognostic factor. However, this study was performed with patients who had been SF for four years, which was different from previous reports [2,5,12,13]. Thus, it was difficult to directly compare the results of previous reports with those of the present study. The dichotomized approach with the threshold of 120 months was chosen subjectively due to the large number of patients with a symptom duration of more than 10 years, according to the author’s experience in clinical practice. Considering that the proportion of referred patients in this study was 58.7% (details not provided) and inadequate control of seizures was a major reason for the transfer of care, the symptom durations were prolonged, which supported the adoption of a 120-month threshold.

According to the results of the multivariate analysis, among LR patients, the relapse risk was highest in the group with onset in their 20s, and the weighted relative risk score was higher in ER patients than in LR patients in every age group. Moreover, the score was highest in ER patients with onset in their 20s, even higher than that of the LR patients with onset in their 30s, which indicated that the overall prognosis was worse for ER patients than LR patients. Higher weighted risk scores and a higher mean multiplicity of prognostic factors were identified in ER patients than in SF and LR patients. These findings showed there were differences based on the relapse pattern. The second most important prognostic factor in LR patients was >10 GTC seizures. Regarding the number of seizures, the patients who reported more than 10 seizures prior to the initiation of therapy had more than double the odds of developing refractory epilepsy [12], and a comparative study with 10 years of follow-up [13] showed a 4.6-fold higher risk of refractory epilepsy in patients with a high initial seizure frequency, supporting the results of this study. Several studies have suggested that there is a relationship between a high initial seizure frequency and poor outcomes [13,20,21,22].

It took approximately one year of empirical treatment to determine whether AED treatment would result in seizure control, especially among non-referred patients, which was supported by the report [18] that 74% of patients responded to treatment within 1 year. Failure to achieve remission within 1 year in ER patients was revealed to be a significant factor predictive of a poor prognosis in this study. The proportion of SF patients who achieved seizure control within one year after the initiation of AED treatment was 59%. The proportions among LR and ER patients were 55.5% and 10.3%, respectively (details not presented), but these patients ultimately experienced relapse after one SF year.

The finding that seizure onset between the ages of 20 and 29 was an independent factor predictive of a poor prognosis is meaningful, and the failure to achieve remission within one year and >10 GTC seizures prior to the initiation of therapy were also found to be significant factors predictive of a poor prognosis in this study. In addition, there were clear differences in the risk of relapse between LR and ER patients. These results in a large study population suggest the potential efficacy of a treatment strategy based on prognostic groups and provide further knowledge of the natural history of epilepsy. This study design might offer a method to overcome the difficulties associated with conducting a double-blind randomized trial. We think that this study design is not a perfect solution but is an alternative approach for studying these issues, especially for the relapse patterns. This study was performed in only one tertiary referral center might limit interpretation of the meaning of the result, even the total enrolled patients were the most up to now. Additional studies are needed to confirm these results.

## Figures and Tables

**Figure 1 diagnostics-10-01089-f001:**
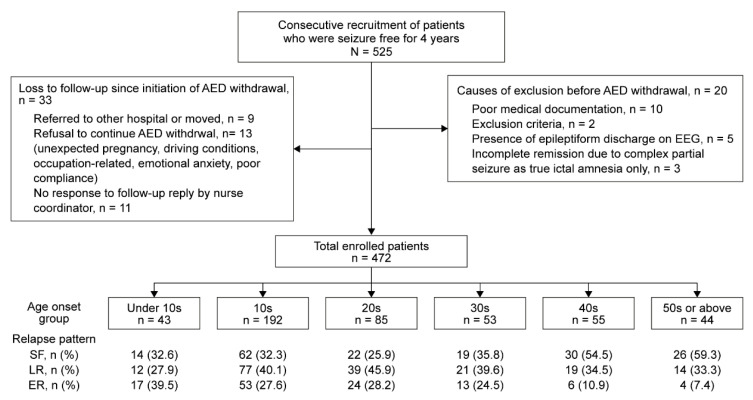
Flowchart of the recruitment process for the inclusion of patients. A total of 525 epilepsy patients were initially recruited, and 53 patients were excluded due to the above documented causes. Finally, 472 patients were grouped into 10-year age groups according to the age of onset and classified as seizure-free (SF), late relapse (LR), and early relapse (ER) patients.

**Figure 2 diagnostics-10-01089-f002:**
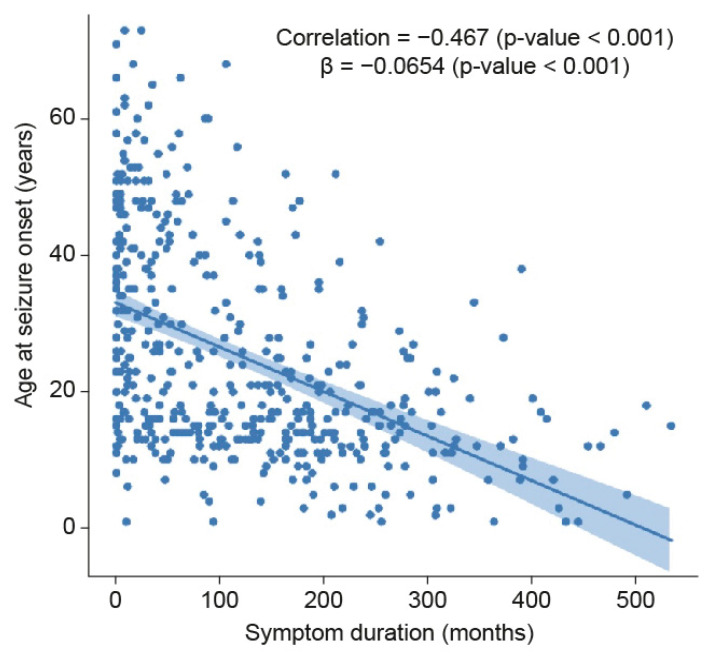
A relationship between age at onset of epilepsy and symptom duration. A statistically significant relationship was noted between age at onset of epilepsy and symptom duration in all subjects.

**Figure 3 diagnostics-10-01089-f003:**
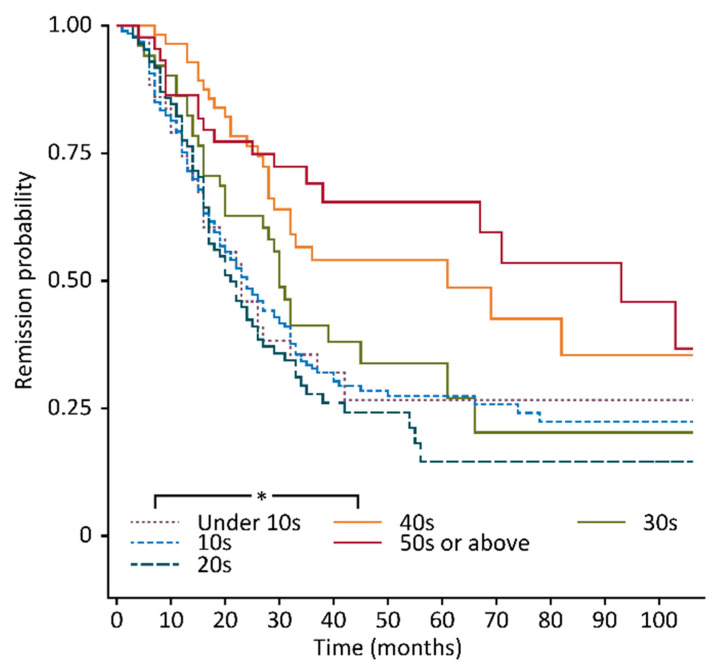
Remission probabilities presented by the Kaplan–Meier curve for each age group. The group with onset in their 20s had the worst remission probability according to the Kaplan–Meier curve, and the groups with onset after age 40 had a significantly better prognosis than those with onset before age 30 (* significant according to the log-rank test with Bonferroni adjustment).

**Figure 4 diagnostics-10-01089-f004:**
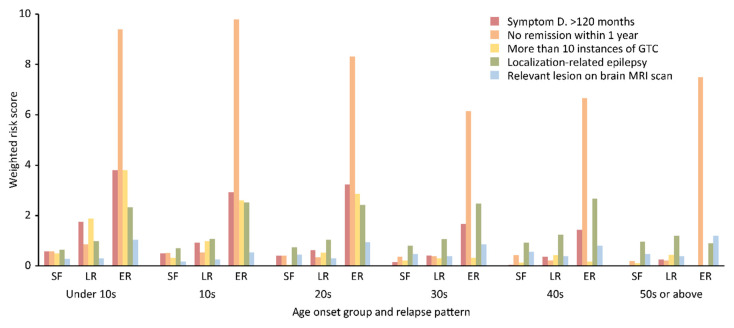
Important prognostic factors in each age group related to the relapse patterns. No remission within 1 year in ER patients was the most important factor in every age group, and localization-related epilepsy in SF and LR patients was the most important factor in every age group, except the group with onset before age 10. The weighted risk scores for almost all prognostic factors in every age group were higher in ER patients than in SF and LR patients.

**Figure 5 diagnostics-10-01089-f005:**
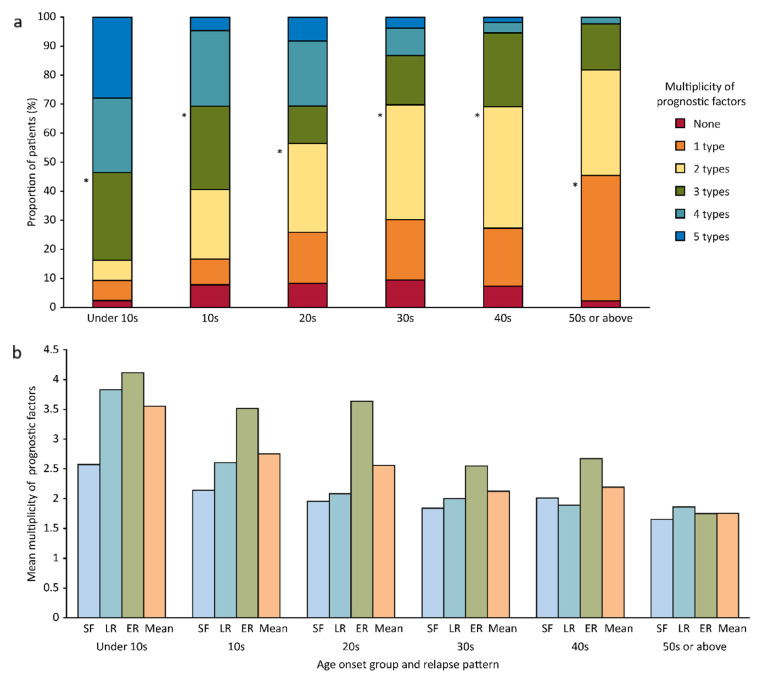
Comparison of the multiplicity (**a**) and mean multiplicity (**b**) of prognostic factors across the age groups stratified by relapse pattern. (**a**) The largest proportion of patients in the group with onset age under 10 years old had three prognostic factors *, namely, a symptom duration >120 months, no remission within 1 year, and >10 GTC seizures. In the group with age of onset between 10 and 19 years old, the factors were a symptom duration >120 months, >10 GTC seizures, and localization-related epilepsy. In groups with onset at 20–29 years old, 30–39 years old, and 40–49 years old, two prognostic factors were present in the greatest proportion of patients, namely, localization-related epilepsy and relevant lesions on brain MRI scans. However, in the group with onset at age 50 or older, localization-related epilepsy was the most important prognostic factor. (**b**) The mean multiplicities of the prognostic factors showed marginal significance with a decreasing tendency with increasing age. Those in LR and ER patients but not in SF patients showed a significant decreasing tendency.

**Table 1 diagnostics-10-01089-t001:** Demographic data and prognostic variables according to relapse pattern in groups stratified by age of onset.

Age of Onset Group, *n*, (%) †	Under 10 Years Old, *n* = 43 (9.1)			10–19 Years Old, *n* = 192 (40.7)			20–29 Years Old, *n* = 85 (18.0)		
Relapse Pattern	SF	LR	ER	SF	LR	ER	SF	LR	ER
*n*, (%) ††	14 (32.6)	12 (27.9)	17 (39.5)	62 (32.3)	77 (40.1)	53 (27.6)	22 (25.9)	39 (45.9)	24 (28.2)
Demographic data									
M: F	9:5	7:5	10:7	35:27	40:37	31:22	15:7	28:11	17:7
Mean age at seizure onset ^a^ ± SD, y	5.0 ± 2.6	5.0 ± 2.7	5.1 ± 3.1	14.1 ± 2.4	14.5 ± 2.6	13.8 ± 2.2	23.8 ± 2.9	23.9 ± 2.9	23.7 ± 2.9
Mean duration of symptoms ^b^ ± SD, m	164.8 ± 141.8	286.7 ± 90.7	252.9 ± 108.0	138.9 ± 117.7	145.9 ± 114.2	183.3 ± 104.3	100.2 ± 95.8	95.6 ± 80.8	179.7 ± 106.6
Prognostic variables, (%) †††									
Antecedents of epilepsy	7 (50.0)	7 (58.3)	10 (58.8)	20 (32.3)	28 (36.4)	23 (43.4)	9 (40.9)	14 (35.9)	12 (50.0)
Symptom duration > 120 m	8 (57.1)	12 (100.0)	15 (88.2)	30 (48.4)	41 (53.2)	36 (67.9)	9 (40.9)	14 (35.9)	18 (75.0)
No remission within 1 y	8 (57.1)	10 (83.3)	16 (94.1)	32 (51.6)	40 (51.9)	52 (98.1)	9 (39.9)	13 (33.3)	20 (83.3)
More than 10 GTC seizures	7 (50.0)	11 (91.7)	15 (88.2)	20 (32.3)	37 (48.1)	32 (60.4)	1 (4.5)	10 (25.6)	16 (86.7)
Localization-related epilepsy	9 (64.3)	9 (75.0)	13 (76.5)	39 (62.9)	60 (77.9)	48 (90.6)	14 (63.6)	31 (79.5)	19 (79.2)
Relevant lesions on brain MRI scan	4 (28.6)	4 (25.0)	11 (64.7)	11 (17.7)	22 (28.6)	18 (34.0)	10 (45.5)	13 (33.3)	14 (58.3)
Nocturnal preponderance	3 (21.4)	6 (50.0)	5 (29.4)	21 (33.9)	24 (31.2)	13 (24.5)	10 (45.5)	14 (35.9)	8 (33.3)

SF—seizure-free; LR- late relapse; ER—early relapse; M—male; F—female; y—year; m—month; GTC—generalized tonic–clonic. ^a^: Significant increase in the mean age at onset of epilepsy with increasing age (*p* < 0.001) and no statistically significant differences among relapse patterns within each age group. ^b^: Significant decrease in the mean symptom duration (*p* < 0.001). †: The proportion of patients in each age of onset group. ††: The proportion of patients with each relapse pattern in each age of onset group. †††: The proportion of patients with prognostic variables according to relapse pattern.

**Table 2 diagnostics-10-01089-t002:** Demographic data and prognostic variables according to relapse pattern in groups stratified by age of onset.

Age of Onset Group, *n*, (%) †	30–39 Years Old, *n* = 53			40–49 Years Old, *n* = 55			50 Years Old or Older, *n* = 44		
Relapse Pattern	SF	LR	ER	SF	LR	ER	SF	LR	ER
*n* (%) ††	19 (35.8)	21 (39.6)	13 (24.5)	30 (54.5)	19 (34.5)	6 (10.9)	26 (59.3)	14 (33.3)	4 (7.4)
Demographic data									
M: F	11:8	13:8	6:7	19:11	5:14	4:6	15:11	6:8	3:1
Mean age at epilepsy onset ^a^ ± SD, y.	34.6 ± 2.8	34.8 ± 3.3	34.1 ± 2.5	44.7 ± 3.5	45.3 ± 2.7	42.5 ± 2.2	58.1 ± 6.3	57.8 ± 7.1	56.8 ± 6.9
Mean duration of symptoms ^b^ ± SD, m.	58.7 ± 72.9	66.6 ± 74.6	124.1 ± 128.4	43.6 ± 52.4	61.7 ± 61.3	61.3 ± 57.4	24.3 ± 27.6	56.2 ± 64.1	39.0 ± 34.6
Prognostic variables (%) †††									
Antecedents of epilepsy	6 (31.6)	9 (42.9)	5 (38.5)	13 (43.3)	6 (31.6)	4 (66.7)	14 (53.8)	6 (42.9)	1 (25.0)
Symptom duration > 120 m	3 (15.8)	5 (23.8)	5 (38.5)	1 (3.3)	4 (21.1)	2 (33.3)	0 (0)	2 (14.3)	0 (0)
No remission within 1 y	7 (26.8)	8 (38.1)	8(61.5)	13 (43.3)	4 (21.1)	4 (66.7)	5(19.3)	3 (11.4)	3 (75.0)
More than 10 GTC seizures	4 (21.1)	3 (14.3)	1 (7.7)	4 (13.3)	4 (21.1)	1 (16.7)	3 (11.5)	3 (21.4)	0 (0)
Localization-related epilepsy	12 (63.2)	17 (81.0)	12 (92.3)	25 (83.3)	16 (84.2)	6 (100)	23 (88.5)	12 (85.7)	1 (25.0)
Relevant lesions on brain MRI scan	9 (47.4)	9 (42.9)	7 (53.8)	17 (56.7)	8 (42.1)	3 (50.0)	12 (46.2)	6 (42.9)	3 (75.0)
Nocturnal preponderance	7 (36.8)	5 (23.8)	3 (23.1)	1 (3.3)	8 (42.1)	0 (0.0)	3 (11.5)	3 (21.4)	1 (25.0)

SF—seizure-free; LR- late relapse; ER—early relapse; M—male; F—female; y—year; m—month; GTC—generalized tonic–clonic. ^a^: Significant increase in the mean age at onset of epilepsy with increasing age (*p* < 0.001) and no statistically significant differences among relapse patterns within each age group. ^b^: Significant decrease in the mean symptom duration (*p* < 0.001). †: The proportion of patients in each age of onset group. ††: The proportion of patients with each relapse pattern in each age of onset group. †††: The proportion of patients with prognostic variables according to relapse pattern.

**Table 3 diagnostics-10-01089-t003:** Prognostic analysis with a univariable multinomial model in relapsed patients with SF patients as the reference.

Variable	Level	VIF	SF Patients (Ref.)						
			LR Patients				ER Patients		
			OR	95% CI	*p*-Value	OR		95% CI	*p*-Value ^a^
Age of onset group	Under 10 years old	2.17	1.59	0.58–4.36	0.366	7.89		2.22–28.05	0.001
	10–19 years old	3.70	2.31	1.11–4.79	0.025	5.56		1.82–16.93	0.003
	20–29 years old	2.58	3.29	1.43–7.58	0.005	7.09		2.13–23.56	0.001
	30–39 years old	1.99	2.05	0.84–5.04	0.117	4.45		1.25–15.79	0.021
	40–49 years old	2.00	1.18	0.49–2.80	0.714	1.30		0.33–5.11	0.708
	50 years old or older		Ref.						
Antecedents of epilepsy	Yes	1.72	0.94	0.62–1.44	0.780	1.34		0.83–2.15	0.229
	No		Ref.						
Symptom duration: 120 months	>120	1.18	1.76	1.13–2.70	0.013	4.31		2.62–7.11	<0.0001
	≤120		Ref.			Ref.			
No remission within 1 year	Yes	1.18	1.02	0.67–1.55	0.9371	9.977		5.288–18.826	<0.0001
	No		Ref.			Ref.			
More than 10 GTC seizures	Yes	1.37	2.05	1.29–3.67	0.003	4.30		2.58–7.15	<0.0001
	No								
Localization-related epilepsy	Yes	1.44	1.31	0.75–2.27	0.340	2.68		1.26–5.70	0.011
	No		Ref.			Ref.			
Relevant lesions on brain MRI scan	Yes	1.12	0.9	0.58–1.39	0.643	1.60		0.99–2.58	0.053
	No								

VIF—variance inflation factor; SF—seizure-free; LR—late relapse; ER—early relapse; OR—odds ratio; CI—confidence interval (between lower 25% and upper 75%); Ref.—reference; GTC—generalized tonic–clonic. ^a^: *p*-value: univariate multinomial regression analysis.

**Table 4 diagnostics-10-01089-t004:** Prognostic multivariable multinomial analysis in relapsed patients with SF patients as the reference.

Covariate	Level	SF Patients (Ref.)					
		LR Patients			ER Patients		
		OR (95%, CI)	*p*-Value	Relative Risk ^a^	OR (95%, CI)	*p*-Value	Relative Risk ^a^
Age of onset group	Under 10 years old	1.17 (0.38–3.63)	0.782	2.18	1.53 (0.34–6.78)	0.580	7.58
	10–19 years old	1.91 (0.86–4.25)	0.112	1.79	1.92 (0.55–6.73)	0.310	7.91
	20–29 years old	3.11 (1.30–7.4)	0.011	3.22	3.42 (0.90–12.94)	0.070	19.44
	30–39 years old	2.12 (0.85–5.32)	0.108	1.44	3.12 (0.79–12.35)	0.106	6.73
	40–49 years old	1.17 (0.48–2.83)	0.732	3.37	0.83 (0.19–3.59)	0.804	12.99
	50 years old or older	Ref.		1.41	Ref.		8.50
Symptom duration: 120 months	>120	1.26 (0.73–2.17)	0.408		1.60 (0.85–3.04)	0.146	
	≤120	Ref.			Ref.		
No remission within 1 year	Yes	0.80 (0.50–1.28)	0.347		7.18 (3.63–14.22)	<0.001	
	No	Ref.			Ref.		
More than 10 GTC seizures	Yes	1.88 (1.09–3.25)	0.024		2.82 (1.49–5.33)	0.001	
	No	Ref.			Ref.		
Localization-related epilepsy	Yes	1.60 (0.89–2.87)	0.119		2.64 (1.14–6.13)	0.023	
	No	Ref.			Ref.		
Relevant lesions on brain MRI scan	Yes	0.87 (0.54–1.40)	0.561		1.83 (1.03–3.27)	0.040	
	No	Ref.			Ref.		

SF—seizure-free; Ref.—reference; LR—late relapse; ER—early relapse; OR—odds ratio; CI—confidence interval; GTC—generalized tonic–clonic. ^a^: Relative weighted risk score obtained by the mean odds ratio of each prognostic factor in each onset age group.

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
