# Peer review of "Prognostic Implications of Epilepsy Onset Age According to Relapse Pattern in Patients with Four-Year Remission"

_diagnostics, 2020, doi:10.3390/diagnostics10121089_

Round 1

Reviewer 1 Report

This is a fine paper presenting several sets of interesting observations. The manuscript needs a careful read through and editing, as it contains multiple typos. This reviewer recommends the study for publication.    

Author Response

As a corresponding author, I earnestly appreciate the editors-in-chief and the reviewers for your dedicated efforts to maintaining the quality of the journal “Diagnostics” and this academic field. I also heartily appreciate the comments from the reviewers, which will surely improve the clarity of the manuscript and upgrade the quality of our work. We did our best to execute the suggestions of the reviewers in revising the manuscript. However, your opinions are still needed for further improvement. Our answers to your comments are as follows.

Reviewer 1 comment

The suggestions about “Research design” and “methods” ; Can be improved

Cf. the number of the line indicates that of the line after revision by authors. The portions edited by authors was brightened as a light blue colored letter and the portions edited by MDPI English editing service to ensure English language accuracy was presented as violet colored.

Response

As you know, a double-blind randomized controlled trial is very difficult in individuals with epilepsy due to confounding sociocultural environmental factors, which has resulted in a wide ranged relapse rate and the proportion of patients experiencing relapse during or after AED withdrawal ranged from 12 to 66 % in literature reviews. That is the reason why author tried to design according to the timing of relapse, which might offer a method of overcoming the difficulties associated with conducting a double-blind randomized trial. Author think this study design is not a perfect solution but an alternative approach to studying these issues, which was published in epilepsia 58;60-67; 2017. One of the results from the previous study was that age at epilepsy onset and symptom duration, as negative prognostic factors, significantly affected the relapse patterns of the late relapse (LR) and early relapse (ER) groups. But the age at onset of epilepsy and symptom duration are closely correlated with each other, as I mentioned that in the beginning of the introduction of this manuscript. So, I tried to assessing these factors in detail according to onset age and also, to clarify the prognostic difference between the ER and LR group.

Actually, this study has been still going on after the end point of analysis of the previous study. Patients’ enrollment and follow-up had been continuously monitored and consecutively recruited. That is the reason why I overlooked the description about the patient recruitment in the methods as both reviewers pointed out. I absolutely agree with the point. We revised the section of “Materials and methods” more faithfully.

Preferentially, author think the relocation of the supplementary figure_S1 as below, which was the schematic presentation of the flow chart of the recruitment process for the inclusion of patients, was adequate to be presented as figure 1 after the line 62 (the line 61 as formatted by editorial office before revision) of the part 2. Materials and methods. The numbers of patients in the line 53 and 54 were corrected adequately in the related sentence. The typos, indicating relapse pattern, located in the bottom area of the far left were corrected as follows; LF-> LR and EF-> ER. These could give the reader more faithful understanding about the recruitment process.

Figure 1. Flow-chart of the recruitment process for the inclusion of patients

(TIF image file could not be presneted here. Please refer to the attached file.

In the attached file, the responses to the reviewer 1. and 2. were included.)

A total of 525 epilepsy patients was initially recruited, and 53 patients were excluded due to the above documented causes. Finally, 472 patients were grouped into 10-year age groups according to the age of onset and classified as SF, LR, and ER patients.

Reviewer 2 Report

The study is well designed and described in the statistical part but has major problems in the recruitment of patients. It  is well known that the major role in epilepsy prognosis is the cause of the epilepsy itself, and this concept led to the new classification of epilepsy syndromes and seizures. From this point of view this study seems a bit "old", with a very limited description of structural or genetic causes related to the age of onset of the epilepsy in the different age groups. Furthermore, it's not correct the exclusion of patients with juvenile myoclonic epilepsy. In this particular epilepsy subset (and in all the genetic generalized epilepsies) the relapse pattern of seizures is still debated and represents a major concern for clinicians and patients.

Author Response

As a corresponding author, I earnestly appreciate the editors-in-chief and the reviewers for your dedicated efforts to maintaining the quality of the journal “Diagnostics” and this academic field. I also heartily appreciate the comments from the reviewers, which will surely improve the clarity of the manuscript and upgrade the quality of our work. We did our best to execute the suggestions of the reviewers in revising the manuscript. However, your opinions are still needed for further improvement. Our answers to your comments are as follows.

Response

As you know, a double-blind randomized controlled trial is very difficult in individuals with epilepsy due to confounding sociocultural environmental factors, which has resulted in a wide ranged relapse rate and the proportion of patients experiencing relapse during or after AED withdrawal ranged from 12 to 66 % in literature reviews. That is the reason why author tried to design according to the timing of relapse, which might offer a method of overcoming the difficulties associated with conducting a double-blind randomized trial. Author think this study design is not a perfect solution but an alternative approach to studying these issues, which was published in epilepsia 58;60-67; 2017. One of the results from the previous study was that age at epilepsy onset and symptom duration, as negative prognostic factors, significantly affected the relapse patterns of the late relapse (LR) and early relapse (ER) groups. But the age at onset of epilepsy and symptom duration are closely correlated with each other, as I mentioned that in the beginning of the introduction of this manuscript. So, I tried to assessing these factors in detail according to onset age and also, to clarify the prognostic difference between the ER and LR group.

Cf. the number of the line indicates that of the line after revision by authors. The portions edited by authors was brightened as a light blue colored letter and the portions edited by MDPI English editing service to ensure English language accuracy was presented as violet colored.

cf. Please refer to the attached revision letter. In the letters, the responses to the reviewer 1. and 2. were included.

Reviewer 2 comments

  1. 1. It is well known that the major role in epilepsy prognosis is the cause of the epilepsy itself, and this concept led to the new classification of epilepsy syndromes and seizures. From this point of view this study seems a bit “old”, with a very limited description of structural or genetic causes related to the age of onset of the epilepsy in the different age groups.

Response:

The authors agree with reviewer’s opinion. We tried to reveal the prognostic impact of the cause of epilepsy. In the table 1: Demographic data and prognostic variables according to the relapse patterns stratified by age onset, those related prognostic variables in the far left column, which were antecedent of epilepsy, localization-related epilepsy, and relevant lesions on brain MRI scan, were corresponding to the causes of epilepsy. In the line from 128 to 133 (line from 114 to 117 of the manuscript, formatted by editorial office), author mentioned about the details of the antecedent as follows; perinatal insults, febrile convulsions, family history of epilepsy, head trauma combined with loss of consciousness for >1 hour, and history of infections of the central nervous system were regarded as antecedents related to epilepsy. The data in the table 1 about localization-related epilepsy and relevant lesions on brain MRI scan could be complementary for a very limited description of structural or genetic causes. The decision of localization-related epilepsy, including remote symptomatic epilepsy in the line 116 and 117 was based on the partial feature of ictal semiology, localized epileptiform discharge in regular follow-up of the electroencephalogram which was performed at intervals at least 2-years, and relevant lesion in a brain MRI scan. So, in this study, cryptogenic epilepsy was included in localization-related epilepsy and remaining patients belonged to idiopathic generalized epilepsy. Patients with undifferentiated epilepsy classification were treated as a missing value statistically, which was not presented in the table for the readability. The results from the logistic regression model were presented in the table 3 and 4.

We added the description in the line of 128 to 133; “was based on the partial feature of ictal semiology, localized epileptiform discharge in the regular follow-up of the electroencephalogram, which was performed at intervals at least 2-years, and relevant lesions in a brain MRI scan, was defined according to the criteria for epilepsy and epileptic syndrome proposed by the International League Against Epilepsy (ILAE) [11]. Cryptogenic epilepsy was included in localization-related epilepsy and undifferentiated patients were treated as missing value statistically” after the phrase: ‘localization-related epilepsy, including remote symptomatic epilepsy in the section of 2.5 Data Analysis)

  1. 2. It is not correct the exclusion of patients with juvenile myoclonic epilepsy. In this particular epilepsy subset (and in all the genetic generalized epilepsies), the relapse pattern of seizures is still debated and represents a major concern for clinicians and patients.

Response:

Author absolutely agree the reviewer’s opinion. Patients with juvenile myoclonic epilepsy (JME) or juvenile absence epilepsy exhibit a greater tendency towards relapse, which need long-term treatment even though seizure control was achieved for a while, compared to the other type of epilepsy. That is the reason why patients with JME have been excluded in a prognosis study of epilepsy. JME was also excluded the inclusion criteria in the previous my report in epilepsia 58;60-67; 2017. We think JME should be treated separately for the prognosis study.

We added the description the reason why JME was excluded in the line 55 and 56 of the section 2.1. Patients Registration and Inclusion Criteria, as follows: patients with juvenile myoclonic epilepsy, “who exhibit a greater tendency towards relapse,“

Finally, if there was no specific limitation of the presenting number of figure in the manuscript, we think figure S2, which represent a significant relationship between age at onset of epilepsy and symptom duration in all subjects, was one of the important results in terms of the purpose of this study. So, we think it is better to be relocated as figure 2 with legend rather than the supplementary figure S2 after the line 168 in the last sentence of 3.1. Demographic Characteristics.

The avove mentioned figure 2. could not be inserted here. Please refer to the attached file.

Round 2

Reviewer 2 Report

I think that the paper is now suitable for pubblication

Author Response

Thank you for your response